# Artificial Cultivation Changes Foliar Endophytic Fungal Community of the Ornamental Plant *Lirianthe delavayi*

**DOI:** 10.3390/microorganisms11030775

**Published:** 2023-03-17

**Authors:** Nan-Nan Wu, Zhao-Ying Zeng, Qin-Bin Xu, Han-Bo Zhang, Tao Xu

**Affiliations:** 1School of Ecology and Environmental Science, Yunnan University, Kunming 650106, China; 2State Key Laboratory for Conservation and Utilization of Bio-Resources in Yunnan, Yunnan University, Kunming 650106, China

**Keywords:** foliar endophytic fungal community, artificial cultivation, wild, ornamental plant, *Lirianthe delavayi*

## Abstract

Many wild ornamental plant species have been introduced to improve the landscape of cities; however, until now, no study has been performed to explore the composition and function of foliar endophytes associated with cultivated rare plants in cities after their introduction. In this study, we collected the leaves of the healthy ornamental plant *Lirianthe delavayi* from wild and artificially cultivated habitats in Yunnan and compared their diversity, species composition, and functional predictions of their foliar endophytic fungal community based on high-throughput sequencing technology. In total, 3125 ASVs of fungi were obtained. The alpha diversity indices of wild *L. delavayi* populations are similar to those of cultivated samples; however, the species compositions of endophytic fungal ASVs were significantly varied in the two habitats. The dominant phylum is Ascomycota, accounting for more than 90% of foliar endophytes in both populations; relatively, artificial cultivation trends to increase the frequency of common phytopathogens of *L. delavayi*, such as *Alternaria*, *Erysiphe*. The relative abundance of 55 functional predictions is different between wild and cultivated *L. delavayi* leaves (*p* < 0.05); in particular, chromosome, purine metabolism, and peptidases are significantly increased in wild samples, while flagellar assembly, bacterial chemotaxis, and fatty acid metabolism are significantly enhanced in cultivated samples. Our results indicated that artificial cultivation can greatly change the foliar endophytic fungal community of *L. delavayi*, which is valuable for understanding the influence of the domestication process on the foliar fungal community associated with rare ornamental plants in urban environments.

## 1. Introduction

Most terrestrial plants interact with fungi, either as endophytic, asymptomatic components living in plant tissues without signs of disease, or as arbuscular mycorrhizal symbiosis with plant roots [1,2]. Endophytic fungi can be symbionts or decomposers; they can also be potential pathogens, and symptoms occur only when the host plant experiences some type of physiological stress [3,4]. After infection with endophytic fungi, some host plants often show the advantages of rapid growth, resistance to adversity and stress, and they are more competitive than uninfected plants, such as through the production of anti-herbivore or anti-microbial toxins, or through the acquisition and biosynthesis of nutritional compounds [5,6]. In general, the endophytes can help host plants adapt to environmental changes [4].

The research suggests that host plant filtering and abiotic environmental variables, e.g., soil (pH, nutrients, and particle size distribution), climate (mean annual temperature (MAT), and mean annual precipitation (MAP)) have been observed to affect the endophytic fungal community [7]. Fungal taxa are known to prefer different niche properties involving nutrients and climate [8,9,10,11,12]. For example, the fungal community composition varied between nutrient-rich and nutrient-poor habitats [8,10]. In addition, some fungi only distributed in habitats with lower MAT and exhibited narrow temperature breadths, while some fungi were detected across wide temperature ranges and in warm habitats [12]. Since the interaction between plants and endophytic fungi represents an important aspect of the plant life cycle, it is very important to understand the endophytic fungal community of rare and endangered plants for the achievement of the conservation goal [13,14].

With the development of cities, all countries have introduced and artificially cultivated wild ornamental plant resources to create the characteristic landscape of cities [15]. These introductions have led to a large number of wild plants being transplanted from the natural environment to the urban environment, accompanied by great changes in plant habitat. Moreover, the excessive use of pesticides and fertilizers will inevitably affect the composition and function of microbial communities in a cultivated plant [16]. Studies have shown that the microbiome can promote the absorption of plant nutrients, improve plant stress resistance and disease resistance, improve soil, and regulate plant immune responses [16,17,18]. In particular, plant leaves support the entire food web of the ecosystem and represent the key compartment niche of the interaction between plants and their microbiota [19]. Therefore, foliar microorganisms, referring to fungi and bacteria living on the surface of (epiphytes) or inside (endophytes) the leaves [20,21], are the most important plant microbial groups, except for root microorganisms [22]. Nonetheless, there are few studies on the foliar endophytic fungi of endangered plants during the domestication process.

*Lirianthe delavayi* belongs to the *Lirianthe* of Magnoliaceae, a dicotyledonous plant class of angiosperm. Its bark and flowers can be used as medicines [23]. As an endemic species in China, *L. delavayi* is only distributed in Southwest China, mainly in the Yunnan Province. *L. delavayi* is one of the most precious tree species cultivated in the subtropical zone. It is a famous garden ornamental tree species, with a tree age of up to 1000 years. When it is planted in the lawn, courtyard, building entrance, and both sides of the boulevard, it can produce a nice background effect and has important ornamental value [24]. It can expect that *L. delavayi* experiences a great change in the foliar endophytic fungal community of *L. delavayi* after artificial cultivation.

In this study, the leaves of wild and artificially cultivated *L. delavayi* are collected in Yunnan, and the foliar endophytic fungal communities in the two habitats are compared by high-throughput sequencing. Because plants can be colonized by endophytic fungi that originate from a neighbor or by spores that persist in the local environment, it is not surprising that endophyte communities found in the same plant host species show striking geographical differences [25]. Moreover, artificial interference, such as pesticides, fertilizers, or planting methods, will also have a certain impact on the phyllospheric microbial community [26]. Therefore, compared with *L. delavayi* population in the wild, we expected that those *L. delavayi* populations in urban ecosystem would have: (i) a reduced diversity and low variation in species composition of the foliar endophytic fungal community; (ii) a high frequency of common fungal pathogens of ornamental plants and commercial crops; and (iii) the increased functions of endophytic fungi for stress resistance.

## 2. Materials and Methods

### 2.1. Study Population and Sample Source

To exclude the climate influence of the microbial community, our comparison study is performed in a small geographic range. *L. delavayi* wild populations mainly distribute in the central Yunnan Province. Therefore, Kunming City, the typical area of the central Yunnan Province, has a long history of artificial cultivation of *L. delavayi*. It is relatively easy to sample both the cultivated populations and wild populations in this area. In November, 2021, we selected nine trees of wild *L. delavayi* populations in three plots in Yiliang County, Yunnan Province. Each three trees are from each plot, with a distance among trees of less than 100 m, and the distance between the three plots is less than 2 km to preserve relatively consistent environmental conditions. Because only eight trees of the cultivated *L. delavayi* populations were found in the campus of Yunnan University, Kunming City, Yunnan Province, all these trees were sampled. Each leaf sample includes three mature and healthy leaves from an individual tree, treated as one biological replicate. In total, we obtained nine biological replicates for the wild population and eight replicates for the cultivated population (Figure 1, Appendix A). The vegetation around the two habitats is quite different. There are diverse wild plants around the wild *L. delavayi* while the plants around the cultivated samples are relatively simple, mainly ornamental shrubs and herbs which are regularly managed, such as daily spraying with pesticides and the pruning of branches and leaves (see Appendix A). A total of 17 samples were collected, as detailed in Appendix A.

The collected *L. delavayi* leaves were pretreated. The healthy and asymptomatic leaves were washed with tap water and dried. After that, the cleaned leaves were surface sterilized: they were first sterilized with 2% sodium hypochlorite solution for 2 min, then sterilized with 75% ethanol solution for 2 min, rinsed with sterile water 3 times, and finally, the surface of the leaves was dried with sterile filter paper, and the leaves were cut into small pieces of 0.1 × 0.1 cm with sterile scissors.

### 2.2. High-Throughput Sequencing and Clustering into Amplified Sequence Variants (ASVs)

Firstly, the CTAB method was selected to extract the total DNA of the microbiome from the leaf samples, and the quality of DNA was detected by agarose gel electrophoresis. At the same time, the DNA was quantified by a UV spectrophotometer. Secondly, the fungal internal transcribed spacer 2 (ITS2) region of the DNA gene was amplified using primers fITS7 and ITS4 [27]. The PCR amplifications were carried out using the following program: 1 min initial denaturation at 94 °C, 19 cycles of 30 s at 94 °C, 30 s at 50 °C, and 45 s at 72 °C for annealing, with a final 10 min elongation at 72 °C. Thirdly, the PCR products were recovered and purified: the PCR products were detected by 2% agarose gel electrophoresis, and the recovery kit used was the AMPure XT beads (Beckman Coulter Genomics, Danvers, MA, USA). Finally, the amplified products were quantitatively mixed and sequenced on the computer: the purified PCR products were evaluated using the Agilent 2100 biological analyzer (Agilent, Santa Clara, CA, USA) and the Illumina library quantitative kit (Kapa Biosciences, Woburn, MA, USA). The qualified library concentration should be more than 2 nM. After gradient dilution of qualified online sequencing libraries (the index sequence is not repeatable), they are mixed according to the required sequencing amount and denatured into a single strand by NaOH for online sequencing, using NovaSeq 6000 sequencer 2 × 250 bp double ended sequencing, and the corresponding reagent is the NovaSeq 6000 SP Reagent Kit (500 cycles).

After the online sequencing was completed, we obtained the original offline raw data, then used overlap to splice the two-terminal data, and performed quality control and chimera filtering to obtain high-quality clean data. DADA2 (Divisive Amplicon Denoising Algorithm) [28] no longer clusters by sequence similarity, but obtains representative sequences with single base accuracy through steps such as “dereplication” (equivalent to clustering by 100% similarity), which greatly improves the data accuracy and species resolution. The core of DADA2 was de-noising, and then the ASVs (amplicon sequence variants) were used to obtain the final ASV table and the numbers of the sequence.

### 2.3. Statistical Analysis

We focused on comparing the differences in the fungal communities between the wild and cultivated trees. Therefore, the sample from each tree was treated as one replicate when performing statistical analysis. All statistical analyses were performed using R-3.4.4, upgma and qiime2 [29]. In order to explore the influence of artificial cultivation on the richness and evenness of the endophytic fungal community in *L. delavayi*, the Shannon index were used to analyze the alpha diversity. In order to find the species diversity difference among the endophytic fungi communities of *L. delavayi* in different habitats, beta diversity analysis was conducted through principal coordinates analysis (PCoA), Adonis analysis, and clustering analysis. The PCoA analysis was based on the Bray–Curtis distance matrix (the most commonly used distance index in the systematic clustering method, which is mainly used to describe the similarity between samples, and the size of the distance is the main basis for sample classification) [30]. Adonis analysis used the Bray–Curtis distance matrix to analyze the explanatory power of different grouping factors. The clustering analysis was based on the unweighted UniFrac and Jaccard distance matrix. At the same time, the UPGMA (unweighted pair group method with arithmetic mean) method was used to cluster the samples to generate clustering tree data files.

Next, for fungal taxon identification, representative sequences for each ASV were blasted against reference sequences in the RDP and Unite databases using BLAST, with a confidence level = 0.8, an identify threshold ≥ 90%, a query coverage ≥ 80%, and an e-value ≤ 10^−5^ [31]. A total of 3125 ASVs were finally obtained, and the taxonomic information of all samples at various levels (phylum, class, order, family, genus, species) was acquired. Endophytic fungal composition differences between the two groups were shown by heat maps. We also explored the influence of artificial cultivation on the endophytic fungal community of *L. delavayi* at the phylum, genus, and species level, and screened the top 100 fungi with *p*-value less than 0.05 (Mann–Whitney U test) for significant difference analysis (basically including all the different fungi).

In addition, in order to explore the changes in the function of endophytic fungi of *L. delavayi* in different habitats, Phylogenetic Investigation of Communities by Reconstruction of Unobserved States (PICRUSt2) software(version picrust2.2.0b) was used to predict the functions of endophytic fungi, and then the obtained function annotation results of the KEGG database were used for STAMP variance analysis [32]. In the KEGG database, biological metabolic pathways are divided into seven categories at level 1, each of which is again systematically classified at level 2, level 3, and level 4, respectively. Level 2 is divided into several sub-paths; level 3 is its metabolic pathway; and level 4 is the specific annotation information of each metabolic pathway map [33,34]. We selected the level 3 metabolic pathways for function annotation in this study, as there are sufficient numbers of categories for analysis and visualization.

## 3. Results

### 3.1. Changes in Diversity of Endophytic Fungi in L. delavayi Leaves

A total of 3125 ASVs were obtained in all samples, and 1617 ASVs were unique to the wild populations, 1005 to the cultivated populations, and 503 were shared by both the wild and cultivated populations (Figure 2a). The Shannon index of wild *L. delavayi* populations is similar to that of the cultivated samples (*p* = 0.77, Figure 2b); and other alpha diversity indices, such as Chao1, observed species, goods_ coverage, Pielou-e, and Simpson, also show the same trends (all *p* > 0.05, Figure 2c and Appendix A). However, PCoA indicated that the endophytic fungal ASVs collected from the two habitats were significantly separated (*p* = 0.001); relatively, the samples from cultivated populations showed less variation than those from the wild group (Figure 2d). Again, cluster analysis also indicated that the composition and structure of the endophytic fungi from the two habitats were quite different (Appendix A).

### 3.2. Endophytic Fungal Species Composition in L. delavayi Leaves

At the phylum level, for the wild populations, 93.43% of the ASVs belong to Ascomycota, 5.20% belong to Basidiomycota, 0.19% belong to Chytridiomycota, less than 0.01% belong to Zygomycota, and 1.19% are unknown; for the cultivated populations on campus, 92.42% of the ASVs belong to Ascomycota, 6.60% belong to Basidiomycota, 0.08% belong to Chytridiomycota, and 0.90% are unknown. No ASV belonging to Zygomycota occurs in the cultivated populations (Figure 3a and Appendix A). At the genus level, the dominant members of both wild and cultivated *L. delavayi* are unclassified. In particular, the abundance of *Sclerostagonospora*, *Erythrobasidium*, and Tremellales_unclassified in the artificially cultivated *L. delavayi* samples is greater than that of wild *L. delavayi*; in contrast, the abundance of *Radulidium*, *Zasmidium*, *Ramichloridium*, Lecanorales_unclassified, Xylariaceae_unclassified, Nectriaceae_unclassified, *Phloeospora*, and Sordariomycetes_unclassified in the wild *L. delavayi* samples is greater than that in the cultivated *L. delavayi* samples (Figure 3b and Appendix A).

We also identified the endophytic fungi that were significantly different between the two habitats at the species level (Figure 4). Eighteen members, including some unclassified species from *Davidiella*, *Kabatiella*, *Didymella*, *Alternaria* etc., and several species of *Mycosphaerella harthensis*, *Mycosphaerella aurantia*, *Sporobolomyces yunnanensis*, *Phoma glomerata*, *Didymella rabiei*, *Aureobasidium pullulans*, *Dioszegia zsoltii*_var_*zsoltii*, *Erysiphe kenjiana* and *Sporobolomyces gracilis*, are enriched in the artificially cultivated *L. delavayi* samples; in contrast, thirteen members, including some unclassified species from *Mycosphaerellaceae*, *Nectriaceae*, *Exobasidium*, etc., and several species of *Phloeospora mimosae-pigrae*, *Radulidium subulatum*, *Zasmidium cellare*, and *Ramichloridium cerophilum* are enriched in the wild *L. delavayi* samples. *Ramichloridium cerophilum* and an unclassified species from *Tilletia* are only detected in the wild *L. delavayi* samples, but *Erysiphe kenjiana* is only detected in the cultivated *L. delavayi*.

### 3.3. Functional Predictions of Endophytic Fungi in L. delavayi Leaves

A total of 296 functions (level 3) were predicted from the endophytic fungi in the *L. delavayi* leaves, mostly related to cell motility, nucleotide metabolism and signal transduction. The relative abundance of the top 55 functional predictions is different between the wild and cultivated *L. delavayi* leaves (*p* < 0.05) (Figure 5); those involving chromosomes, purine metabolism, and peptidases are significantly higher in the wild samples than in the cultivated samples, but those involving flagellar assembly, bacterial chemotaxis, and fatty acid metabolism are significantly higher in the cultivated samples than in the wild samples (*p* < 0.05).

## 4. Discussion

### 4.1. Impact of Artificial Cultivation on the Diversity of Endophytic Fungi in L. delavayi Leaves

This study is the first to determine the impacts of artificial cultivation on the foliar endophytic fungi of *L. delavayi*. Different from our first hypothesis, we found that the endophytic fungi alpha diversity from two *L. delavayi* populations is similar (Figure 2b,c). Similarly, Hassani et al. [35] studied the composition of the microbial community in the leaves of one type of cultivated wheat and two kinds of wild wheat (*Triticum boeoticum* and *T. urartu*), and showed that the diversity of the fungal community showed no significant difference among the three kinds of wheat. In contrast, most studies have previously shown that domestication leads to reduced microbial diversity [36,37], but there are also counterexamples, such as *Phaseolus vulgaris*. Pérez-Jaramillo et al. [38] found that the rhizosphere microbial diversity of modern *P. vulgaris* cultivated in agricultural soil was significantly higher than that of wild *P. vulgaris.* Thus, the effect of artificial cultivation on microbial diversity varies among host plant species, and may be related to the physiological and ecological characteristics and domestication direction [39].

Nonetheless, the source of the phyllospheric microbial community is mainly from air, seeds, soil, and other neighboring plants [40]. Considering the obvious changes in habitat where ornamental plants are transplanted from the wild, it is reasonable to believe that artificial cultivation exerts a profound impact on fungal species composition in the leaves. Accordingly, PCoA analysis indicated that the endophytic fungi community of *L. delavayi* from two habitats was very distinct in species composition; moreover, the variations in ASVs associated with wild *L. delavayi* were more extensive than those of the cultivated group (Figure 2d). This reflects the great environmental heterogeneity of wild *L. delavayi*.

### 4.2. Impact of Artificial Cultivation on the Species Composition of Endophytic Fungi in L. delavayi Leaves

The dominant phylum of *L. delavayi* in two habitats is Ascomycota, accounting for more than 90% of the total ASVs (Figure 3a), similar to previous reports in many plants, such as *Bamboo* [41], *Sinopodophyllum hexandrum* [42], *Pennisetum sinese* [43], and *Eucommia ulmoides* [44]. In contrast to wild *L. delavayi*, however, there was no Zygomycota observed in cultivated *L. delavayi* leaves (Figure 3a). The abundance of Zygomycota is greatly affected by factors such as climate and vegetation [45]; in particular, the diverse plant resources may increase the abundance of Zygomycota [46,47]. Therefore, the low plant resources surrounding the cultivated *L. delavayi* may account for the lack of Zygomycota. At the genus level, *Alternaria* is more abundant in cultivated than in wild *L. delavayi* (Figure 3b). Many members of *Alternaria* are important plant pathogens [48], causing serious leaf spot disease in many ornamental plants and commercial crops, such as *Chrysanthemum* [49], *Cruciferae* [50], *Ophiopogon japonicus* [51], and pear [52]. Moreover, *Alternaria* can disperse by releasing large numbers of airborne spores [53]. In this case, we assumed that many ornamental plants and commercial crops such as *Ophiopogon japonicus* and pear surrounding the cultivated *L. delavayi*, may be as vectors and lead to the high abundance of *Alternaria* in the leaves of the campus *L. delavayi*. Meanwhile, we found a high occurrence of *Erysiphe kenjiana* in leaves of the cultivated *L. delavayi* (Figure 4). *Erysiphe* is also an important plant pathogen, and can cause the disease called Mulberry powdery mildew [54,55,56]. Mulberry powdery mildew is very common in many cultivated ornamental plants, such as *Paeonia lactiflora*, *Tarenaya hassleriana* (Chodat) Iltis, *Symphyotrichum novi-belgii* (L.) G.L.Nesom, and *Calendula officinalis* L. [57], as well as *Hydrangea macrophylla* [58], *Rose* [59], and *Petunia* [60]. There are many ornamental plants such as *Hydrangea macrophylla* and *Rose* on the campus, which may be responsible for the high occurrence of *E. kenjiana*. In the future, it will be necessary to clarify if these surrounding ornamental plants indeed cause a high occurrence of these potential pathogens in the artificially cultivated *L. delavayi*.

Interestingly, we found that the artificial cultivation obviously reduced the abundance of members belonging to Mycosphaerellaceae in *L. delavayi* leaves (Figure 4). Many species in the family Mycosphaerellaceae are related to a variety of plant diseases [61], which mainly damage plant leaves and affect the growth, yield, and quality of plants [62]. The reduced abundance of members from Mycosphaerellaceae in cultivated *L. delavayi* leaves may reflect the fact that cultivated *L. delavayi* has an adverse effect on the growth of these fungi, physiologically or metabolically, or the increased fungal groups, including *Alternaria* and *Erysiphe*, may compete with it. Previously, white leaf spot (*Neopseudocercosporella capsellae*, Mycosphaerellaceae), and Alternaria leaf spot (*Alternaria brassicae*), as well as other fungal pathogenic species, commonly co-occur in rapeseed (*Brassica napus*). In controlled environment studies, Alternaria leaf spot was reduced on cultivar Thunder TT of *B. napus* when *A. brassicae* was applied following *N. capsellae* inoculation, suggesting an antagonistic interaction between these two pathogens [63].

### 4.3. Effect of Artificial Cultivation on the Functions of Endophytic Fungi Community in L. delavayi Leaves

Besides the changes in species composition, artificial cultivation also enhanced the microbial functions, including chemotaxis and fatty acid metabolism, in the leaves, (Figure 5). It is well known that chemotaxis plays an important role in the microbial disease infection of plants [64]. Moreover, the levels of the unsaturated fatty acids (those that carry double bonds between carbons) 18:1, 18:2, and 18:3 have been verified to be particularly important in plant defense [65]. For example, the overexpression of a yeast ∆9 desaturase increases the accumulation of 16:1 in transgenic tomato and eggplants, and this is associated with increased resistance to powdery mildew [66] and *Verticillium dahliae*, respectively [67]. This suggests a potential role for 16:1 in defense against fungal pathogens. Likewise, increased levels of 18:2 and 18:3 result in higher resistance to *Colletotrichum gloeosporioides* in avocado and *Pseudomonas syringae* in tomato [68,69]. Rhizobacteria-induced enhanced resistance to *Botrytis cinerea* is also associated with the accumulation of 18:2 and 18:3 fatty acids in bean plants [70]. However, it remains to be determined if these increased functions in the cultivated *L. delavayi* are related to the heavy load of potential pathogens, including *Alternaria*, *Erysiphe* (Figure 4). Interestingly, some functions for improving plant growth, including purine metabolism [71] and processing peptidase [72], are more activated in the wild *L. delavayi* samples than in the cultivated samples (Figure 5), inferring that wild *L. delavayi* grows better and stronger than the cultivated populations, a situation observed in the our sampling process (Figure 1). Our predicted functions suggested that the reduced pathogen load of *L. delavayi* may impact the resource allocation of the host. A similar phenomenon has been commonly observed for invasive plants. For example, to grow faster in the introduced range, the invasive plant *Ageratina adenophora* allocates more resources, primarily N, to photosynthesis rather than to defense (cell walls) [73].

In conclusion, our results indicated that artificial cultivation can significantly change the species composition of the foliar endophytic fungi of *L. delavayi*; in particular, artificial cultivation trends to increases the frequency of common phytopathogens of crops, such as *Alternaria*, *Erysiphe*. Over a long period of time, many wild ornamental plant species have been introduced to improve the landscapes of cities, and our data is important for understanding the influence of the domestication process on the endophytic fungal community associated with rare ornamental plants in urban environments.

## Figures and Tables

**Figure 1 microorganisms-11-00775-f001:**
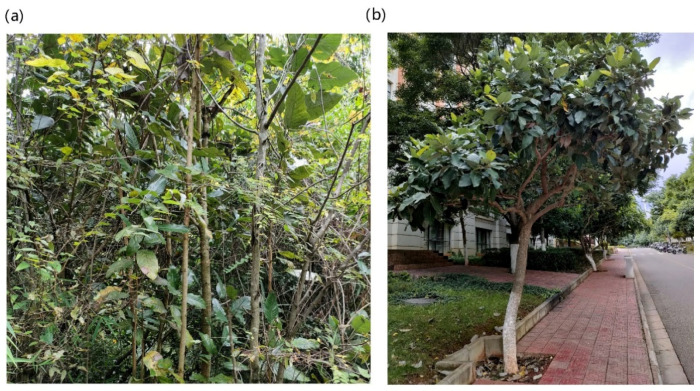
Habitats for representative *L. delavayi* individuals grown in the wild (**a**) and on campus (**b**) in Yunnan Province. The GPS coordinates are given in Appendix A.

**Figure 2 microorganisms-11-00775-f002:**
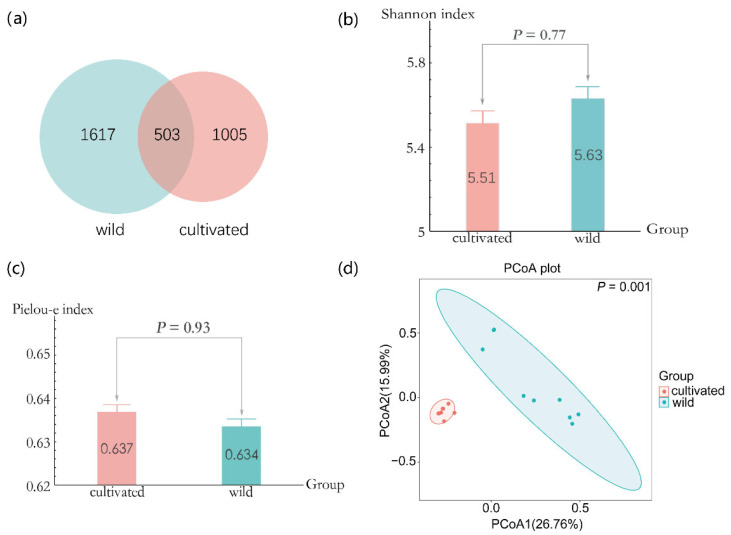
Alpha diversity and beta diversity of *L. delavayi* leaves. (**a**) Venn diagram of endophytic fungi in wild and cultivated *L. delavayi* leaves. (**b**) The Shannon index of the fungal community for wild and cultivated *L. delavayi* leaves. (**c**) The Pielou-e index of fungal community for wild and cultivated *L. delavayi* leaves. (**d**) PCoA analysis of endophytic fungal community in wild and cultivated *L. delavayi*. The percentage of the horizontal and vertical coordinates represents the degree of interpretation of the sample difference for the first axis and the second axis, respectively. The analysis is based on Bray–Curtis distance matrix.

**Figure 3 microorganisms-11-00775-f003:**
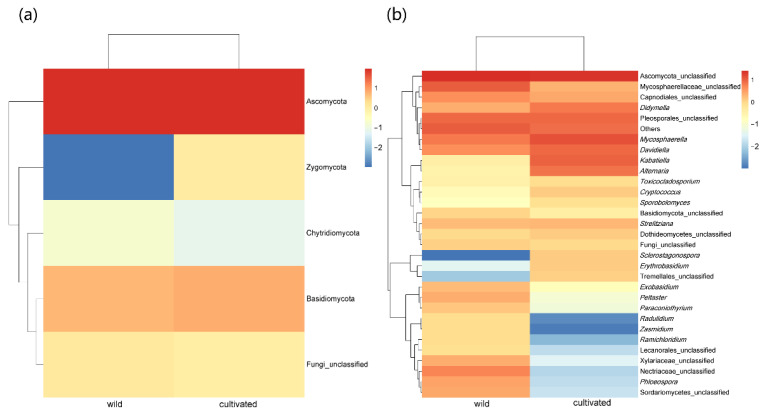
Heat maps of endophytic fungi in wild and cultivated *L. delavayi* leaves at the phylum level (**a**) and the genus level (**b**). In the figure, the gradient from blue to red represents the change of abundance from low to high. After Z-value transformation, the heat map normalizes the expression abundance of the same fungi.

**Figure 4 microorganisms-11-00775-f004:**
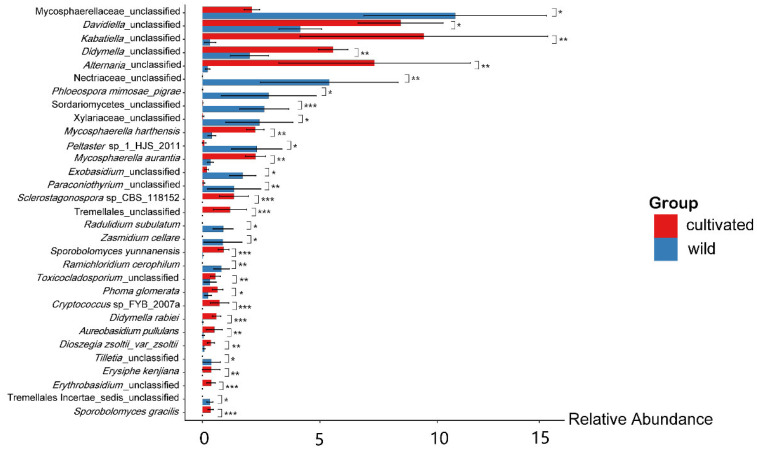
Barplot difference analysis of endophytic fungi at species level in wild *L. delavayi* and cultivated *L. delavayi* leaves. All species were analyzed for differences, and the species with *p*-value less than 0.05 (Wilcoxan test) were screened to draw a histogram (basically including all differential species). In the figure, * represents 0.01 < *p* < 0.05, ** represents 0.001 < *p* < 0.01, and *** represents *p* < 0.001. The ordinate in the figure represents the different species (arranged from left to right according to the abundance), and the abscissa is the relative abundance.

**Figure 5 microorganisms-11-00775-f005:**
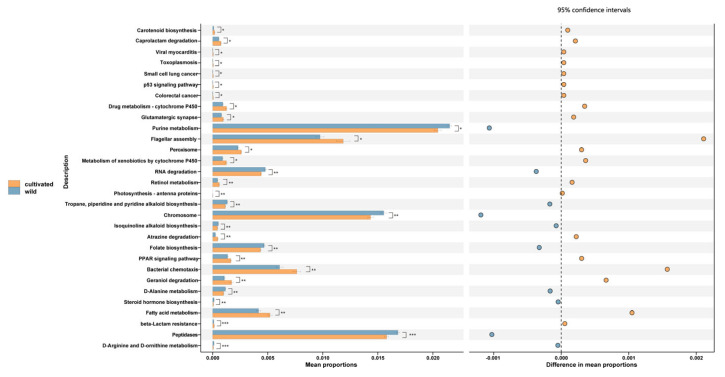
STAMP difference analysis of functional predictions of endophytic fungi in wild and cultivated *L. delavayi* leaves based on the KEGG database (level 3). The STAMP analysis results in the figure only apply to the first 30 functions with *p* < 0.05 in the t−test difference test results of two comparisons. In the figure, * represents 0.01 < *p* < 0.05, ** represents 0.001 < *p* < 0.01, and *** represents *p* < 0.001. The results showed statistically significant functions (95% confidence intervals).

## Data Availability

The datasets presented in this study can be found in online repositories. The names of the repository/repositories and accession number(s) can be found below: https://www.ncbi.nlm.nih.gov/sra/PRJNA890451 (URL (accessed on 14 October 2022)).

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
