# Peer review of "Artificial Cultivation Changes Foliar Endophytic Fungal Community of the Ornamental Plant Lirianthe delavayi"

_microorganisms, 2023, doi:10.3390/microorganisms11030775_

Round 1

Reviewer 2 Report

My comments on reviewing the manuscript titled: Artificial cultivation changes foliar endophytic fungi of the ornamental plant Lirianthe delavayi are :

The study has been conceptualised to detect differences between endophytic populations of fungi in leaves of ornamental plants grown under cultivation and in the wild.

The work has been done well with due care to have replicates that allowed statistical analyses. And the results show significant differences between the fungal microbiomes of the two groups.

The Discussion is relatively short and to the point.

The authors area asked to indicate the source of chemicals and reagents, For example if know the source of the AMPure beads used we could figure out whether they are XT or XP.

The language used is very good except for a few instances which need to be rephrased to get the correct meaning.

For example, the highlighted text needs to be corrected:

Page 2, Last Para: a number of minor grammatical errors

For cultivated population, only eight cultivated L. delavayi trees were found in the campus of Yunnan University, Kunming City, Yunnan Province. Therefore, only eight samples from such eight trees were collected.

Page 3, Para 2: finally sucked the leaves surface moisture with sterile filter paper.

Para 4; After the on-line sequencing was completed, off-line raw data were used overlap to splice the two-terminal data, and performed quality control and chimera filtering to obtain high-quality Clean Data.

Round 2

Reviewer 1 Report

Appreciate the authors efforts to consider the suggestions and improve the quality of a well-designed study. 

Reviewer 2 Report

All the corrections were made to my satisfaction except the last correction- Please change it to; 

we obtained the original off-line Raw Data, then used the overlap to splice the data from the two termini.